# Vaccine Hesitancy in Sub-Saharan Africa in the Context of COVID-19 Vaccination Exercise: A Systematic Review

**DOI:** 10.3390/diseases11010032

**Published:** 2023-02-09

**Authors:** Elizabeth A. Ochola

**Affiliations:** 1Centre for Global Health Research, Kenya Medical Research Institute (KEMRI), P.O. Box 1578-40100, Kisumu 40100, Kenya; eochola@uwaterloo.ca; 2Associate Researcher, School of Bioethics, Av. Universidad Anáhuac 46, Lomas Anahuac, Naucalpan de Juárez 52786, Mexico

**Keywords:** COVID-19, infectious diseases, pandemic, vaccines, vaccine hesitancy, sub-Saharan Africa (SSA), utilitarianism, bioethics

## Abstract

Presently, the COVID-19 vaccine is seen as a means to an end in light of other challenges, such as vaccine inequity. Through COVID-19 Vaccines Global Access (COVAX), an initiative founded to guarantee fair and equitable distribution, vaccine hesitancy remains a critical component that needs to be addressed in sub-Saharan Africa. Utilizing a documentary search strategy and using the keywords and subject headings Utilitarianism and COVID-19 or Vaccine hesitancy and sub-Saharan Africa, this paper identified 67 publications from different databases (PubMed, Scopus and Web of Science), which were further screened by title and full text to achieve (n = 6) publications that were analyzed. The reviewed papers demonstrate that vaccine hesitancy occurs against a colonial backdrop of inequities in global health research, social–cultural complexities, poor community involvement and public distrust. All of these factors undermine the confidence that is crucial for sustaining collective immunity in vaccine programs. Even though mass vaccination programs are known to limit personal freedom, the exchange of information between healthcare professionals and citizens must be improved to encourage complete disclosure of vaccine information at the point of delivery. Moreover, addressing components of vaccine hesitancy should involve relying not on coercive public policies but on consistent ethical strategies that go beyond current healthcare ethics toward broader bioethics.

## 1. Introduction

Infectious diseases such as the influenza pandemic, the Ebola epidemic, and other outbreaks have historical, geographical and socioeconomic implications that have become difficult to address even though vaccines are a sure mode of infection control. It is argued that individuals who have access to vaccines and who do not suffer from the adverse effects of vaccinations have a moral duty to be vaccinated and contribute to herd immunity. It is further asserted that the moral obligation to be vaccinated strengthens vaccination policies. 

The ethical theory of utilitarianism is based on maximizing good, meaning producing the most good for the most people who are equally affected. Furthermore, utilitarianism strives to balance good over harmful consequences by focusing on society rather than individuals [1]. There are two types of utilitarianism, namely act and rule utilitarianism. Act utilitarianism supports an action if it produces the most good for the most people involved, while rule utilitarianism is societal and more rigid with rules considered in sets referred to as moral codes; thus, an optimal code maximizes good for most people. In essence, utilitarianism recognizes that adhering to moral rules ultimately leads to the most good [1].

Herd immunity is a collective and a public good since it involves the cooperative action of people [2]; it benefits all people and ensures fewer resources are directed to caring for the sick. Governments have the imperative to protect the wellbeing of the population and, notably, the public good. The utilitarian argument grounds an individual’s moral obligation to be vaccinated [3] unless the cost of being vaccinated outweighs the benefits. However, vaccination decisions and policies present tension at the intersection of individual rights and moral duty to prevent harm [4], in addition to an increase in the conscientious decision to object to vaccines.

An ethically framed vaccination policy is essential in a society since it provides fairness, democracy and respect for individuals, which are predicted to increase trust and vaccine coverage [5]. Thus, ethical considerations are crucial during vaccine rollout. During vaccinations, some individuals may refuse or stall during vaccine uptake in a process referred to as vaccine hesitancy. By definition, vaccine hesitancy can be broadly defined as a delay in accepting or refusing vaccines despite the availability of the service [6]. However, the World Health Organization (WHO) Strategic Advisory Group on Experts (SAGE) explicitly defines vaccine hesitancy as a delay in accepting or refusing vaccinations despite the availability of vaccination services. This definition reveals that vaccine hesitancy is a complex and context-specific construct that varies across time, place and vaccines. Moreover, vaccine hesitancy is influenced by complacency, convenience and confidence [7].

Most often, vaccine hesitancy is specific to subgroups and populations, and it is characterized by long-standing anxieties toward vaccine uptake due to reasons such as trust, perceived safety, concerns over restrictive legislation that may seek to mandate vaccination coverage, the motive of the pharmaceutical company that is developing the new vaccines and perceptions of professional expertise and authority [8,9]. As such, vaccine hesitancy is a growing problem in public health, with efforts to increase vaccination coverage being met with a subsequent fall in vaccine uptake.

Vaccine skeptics have been present since the smallpox vaccination exercise carried out in the 19th century. The arguments put forward have been more or less consistent, but with evolving media communications [10,11], the concerns have shifted diplomatically. However, the lack of trust in vaccines has led to less protection, more frequent disease outbreaks and a constant threat to herd immunity [5].

The skepticism toward, refusal of and lack of trust in vaccines can be attributed to social, political or safety-related assumptions and concerns indicative of a complex issue [8]. Moreover, some concerns are perennial, while others may be new as each vaccine is introduced in a different context. On investigation, some concerns may be unfounded, while others may be valid [5]. Through research, social scientists have questioned various anti-vaccination movements that demonstrate that public concerns are not merely a distrust of science but a mix of scientific, psychological, sociocultural and political factors [8,12].

The outbreak of coronavirus disease (COVID-19) led to high morbidity and mortality across the world. Recently, the COVID-19 vaccine was rolled out in many countries. However, there have been reported incidences of vaccine hesitancy in many parts of the world, including in sub-Saharan Africa (SSA), which has a history of hesitation, especially when it comes to the introduction of new vaccines [13]. Therefore, the utility of COVID-19 vaccination campaigns is not solely dependent on vaccine efficacy and safety but also on vaccine acceptance for the successful control of the pandemic [14].

Rus and Groselj [15] argue that when individuals lack the moral obligation to contribute to herd immunity, having mandatory vaccination policies is ethically justified since states have the responsibility to protect herd immunity as a common good. For example, if a COVID-19 vaccine was available at the onset of the pandemic to people worldwide, we would not have experienced lengthy lockdowns, economic decline or compromised physical and mental wellbeing. Moreover, fewer people would have died [4]. Currently, the COVID-19 vaccine is seen as a means to an end, even though there are challenges such as vaccine inequity, adverse side effects and hesitancy, among others. Based on these observations, vaccination policies continue to have ripple effects on individuals and communities.

Countries in SSA have faced challenges in COVID-19 vaccine administration. Some of these challenges include poor infrastructure for distributing the vaccine, limited cold chain facilities and growing vaccine hesitancy [16]. At the moment, only 26.8% of the population in SSA is fully vaccinated, compared with more than 50% of the population in the Global North [17]. Platforms such as COVID-19 Vaccines Global Access (COVAX), an initiative spearheaded by WHO alongside other partners, has tried to ensure equitable access to vaccines globally [18]. Even though COVAX was mainly founded on a mission to accelerate the development and manufacture of COVID-19 vaccines by guaranteeing a fair and equitable distribution, vaccine hesitancy [18] remains a critical component of vaccine rollout that needs to be addressed in SSA. As such, this review article evaluates vaccine hesitancy in sub-Saharan Africa in the context of COVID-19 vaccination.

## 2. Methods

This systematic review followed the Preferred Reporting Items for Systematic Reviews and Meta-Analysis (PRISMA) guidelines and is being reviewed for registration into PROSPERO. The documentary search strategy involved keywords and subject headings Utilitarianism and COVID-19 or vaccine hesitancy and sub-Saharan Africa. A review of the journal titles and abstracts was performed to establish a match within the selection criteria. Based on the parameters of interest, this paper included publications that explicitly mentioned vaccines or vaccine hesitancy in the COVID-19 pandemic. The publications that fit the inclusion criteria were obtained for a full-text review. The review excluded publications that reported any form of vaccine hesitancy before the year 2000. This paper utilized a data analysis approach to obtain key details and develop themes for analysis. The data extraction approach took into consideration the publications and database search, study selection criteria, information retrieval and major outcomes (Table 1). The approach was aimed at obtaining key details and developing themes for analysis.

The data extraction approach resulted in identifying 67 publications from the databases that were searched (PubMed, Scopus and Web of Science). Out of the 67 publications, 28 publications were duplicates (when screened by title), hence the remaining 39 publications were considered for screening based on title and abstract. After the screening process by title and abstract, 24 publications were excluded as they were not full texts (they were abstracts only), leaving a final pool of 15 publications that matched the keywords and date of publication and were full texts (eligible). A further 9 publications were excluded as they fell outside the scope of the main theme of the review and the remaining 6 publications were analyzed (Figure 1).

## 3. Results

The results of the reviewed papers demonstrate various levels of vaccine hesitancy which risk undermining the trust that is necessary for sustaining herd immunity in vaccine programs. The first paper, by Afolabi and Ilesanmi [13], reports that COVID-19 vaccine hesitancy in Africa is fueled by public distrust. The poor handling of the pandemic by the government led to distrust from the general public; as such, there was a delayed response to preventive activities in the continent. A laxity in border closures also propelled the importation of the virus to Africa. Other factors included a lack of community involvement in the contexts of social distancing, handwashing and masking, among other measures. The article further argues that African governments may have done little to debunk social and traditional media theories that the African continent is “immune” to the virus due to the tropical climatic conditions. As a result, many Africans lacked confidence in how the virus was reported and handled in the continent. Ekwebelem et al. [19] concur with the WHO 2019 report that vaccine hesitancy is one of the top threats to health and wellbeing. Sub-Saharan Africa is a multicultural and diverse continent; therefore, vaccine hesitancy is driven by various cultural, social, historical, political and individual factors such as values, risk perceptions, emotions, knowledge or beliefs. The social–cultural complexity of the continent has contributed immensely to sporadic vaccine hesitancy, with the hesitation toward COVID-19 vaccines expected to ultimately vary in different contexts, contributing potential hindrances to the vaccine rollout. Furthermore, there are fears, misinformation and conspiracy theories being spread by social influencers, religious leaders, anti-vaccinists and political leaders that the vaccine has the potential to reduce the rising population and track people’s lives through microchips.

The paper by Flint [20] discusses the historical origins of vaccine hesitancy, which stems from a multitude of issues such as colonial geography, the structural adjustment programs of the 1980s and 1990s, the HIV/AIDs pandemic, clinical trials involving pharmaceutical companies and the influence of the global health field. The paper argues that it is important to consider the backdrop of colonialism in understanding hesitancy toward COVID-19 vaccine trials and rethinking more equitable relations within global health. Furthermore, Harrison and Wu [21] discuss the role of public health experts in solving COVID-19 vaccine hesitancy/refusal, with a push for vaccine confidence being seen as a means of conceptualizing and responding to the COVID-19 pandemic. As such, the paper calls for a reimagination of the culture of public health and vaccine safety regulations to spark public confidence in vaccine programs depending on the work performed for the community in social, biological, political and moral contexts. 

Nihlén [5] analyzes vaccination policy from an ethical perspective against the backdrop of growing vaccine hesitancy. Through an ethical lens, the paper analyzes risk communication using examples from the measles and H1N1 vaccination programs and their associated side effects, which propelled hesitancy. Furthermore, Nihlén [5] argues that vaccine skeptics should not be treated as ill-informed or less educated, but their concerns should be addressed respectfully. In addition, the public should trust the message and count on the government to take responsibility for individuals affected by vaccine side effects. Trogen and Pirofski [22] affirm that at the onset of the pandemic, there were initial concerns about the scarcity of COVID-19 vaccines due to an increase in public demand, but as vaccine supply meets demand, vaccine hesitancy is quickly becoming a defining theme of the COVID-19 pandemic. Furthermore, the authors differentiate vaccine hesitancy from vaccine refusal by defining vaccine refusal as carrying with it deep political, cultural and emotional underpinnings that are difficult to overcome, with individuals in this group being labelled as “anti-vaxxers”.

## 4. Discussion

This paper aimed to investigate vaccine hesitancy in sub-Saharan Africa in the context of the COVID-19 vaccination exercise using utilitarianism, which is an ethical theory that prescribes the maximization of a common good. As such, utilitarianism promotes collective good while preserving as much freedom as possible. In this case, an effective rollout of COVID-19 vaccinations and acceptance offers the most promising protection from severe disease, which enhances wellbeing and maximizes utility. Moreover, utilitarianism provides an outlook broader than the individual toward the collective, in this case, the community; thus, public health prevention of COVID-19 is a utilitarian goal since it prevents the public transmission of the virus. Therefore, individuals have a moral obligation toward the collective for a positive outcome [23]; the more people are immunized, the greater the collective benefit of the vaccine [24].

The reviewed papers demonstrate that vaccine hesitancy in SSA occurs against a backdrop of colonialism and inequities in global health research, social–cultural complexities, poor government response in debunking social and traditional media theories and poor community involvement in public health measures. In addition, there is public distrust resulting from delayed responses to border closures and the importation of the virus to Africa; conspiracy theories advanced by social influencers, religious leaders, anti-vaccinists and political leaders; and the anticipated and observed vaccine side effects. All of these factors undermine the confidence that is crucial for sustaining collective immunity in vaccine programs.

This systematic analysis found that the utilitarian goal of promoting public health practices during a vaccination process considers the ethical concepts of personal liberty, freedom of consciousness and the right to autonomy [5,12]. The human body is the sacred and inviolable property of a person; therefore, no one is mandated to participate in preventive or curative treatment without their consent [12]. Even though vaccination is framed as a collective duty in which citizens of welfare states contribute to population health as a measure of good citizenship [12], the right to autonomy must also be considered. Although public health programs are argued to limit freedom, a patient-centered approach is important for trust and decision making. Therefore, efforts to address vaccine hesitancy should not rely on coercive public policies; instead, they should focus on citizen engagement to inform a patient-centered approach and cultivate an ethically consistent strategy [9], hence the need to strengthen the relationship between individual, collective and institutional responsibility to prevent vaccine hesitancy and promote herd immunity.

In the quest to remedy vaccine hesitancy, it has been earlier suggested that governments communicate vaccine risks and benefits in a responsible manner and take responsibility for individuals negatively affected by the adverse effects of the vaccines [5]. Furthermore, vaccine skeptics should not be treated as ill-informed or less educated, but their concerns should be addressed respectfully. In the event that individuals suffer from the side effects of vaccines, the public should be able to count on the governments [5] to responsibly remedy the concern and restore trust. Finally, the citizens of SSA need to be actively involved in the structure and modes of delivery of the COVID-19 vaccine. Additionally, the stakeholders involved in vaccine rollout should acknowledge community efforts in vaccine acceptance and determine areas that require improvement to maintain vaccine acceptance [13].

## 5. Limitations

This systematic review identified 67 publications, but only 6 were included for the analysis of COVID-19 vaccine hesitancy in SSA. The other limitations that were encountered during the systematic review include the following: First, at the onset of the pandemic, the low-resourced nature of many African countries made it challenging to report on the current state of COVID-19 vaccines in the continent. Second, there was the challenge of finding literature that solely tackles religious, cultural and other context-specific concerns relating to vaccine information trends in the African continent. Third, certain persistent challenges in vaccine rollout fall within narrow and clinically oriented modes of thinking. Fourth, vaccine hesitancy is not a new phenomenon, but the current arguments that vaccine skeptics have raised have remained more or less consistent in light of recent developments, as portrayed in traditional and contemporary media. Fifth, COVID-19 vaccines have been shown to have benefits that outweigh the risks; however, studies that contextualize this phenomenon as it relates to vaccine hesitancy in SSA are still few in number.

## 6. Conclusions

Vaccine hesitancy is a growing problem that threatens vaccination uptake; thus, it is essential that governments and other stakeholders understand why individuals are hesitant and what actions can be undertaken to minimize hesitancy. First, addressing vaccine hesitancy requires improving the exchange of information between healthcare professionals and citizens, complete with full disclosure of vaccination information at the point of delivery. Furthermore, addressing vaccine hesitancy should involve relying not on coercive public policies but on an ethically consistent strategy that goes beyond current health care ethics and toward broader bioethics. In doing so, the citizens of SSA will have the right to education, the right to make informed decisions for themselves and the right to support and advise persons who may hesitate to take vaccines. Second, the misconceptions around the COVID-19 vaccine hinder the anticipated success of vaccine rollout. Therefore, response strategies need to be adopted to address misconceptions, and all potential hindrances to vaccine acceptance need to be proactively addressed in a culturally and language-sensitive manner that involves sociocultural influencers and engages community leaders. 

Third, identifying previous biomedical histories could help to avoid past problems related to clinical trials and vaccine rollout as this may contribute to rethinking and creating more equitable relations in terms of global health. Fourth, there is a need to achieve vaccine confidence through public health, which entails having programs broader than the delivery of the vaccine technology. Moreover, developing a COVID-19 vaccine should not be the sole indicator of a successful response nor an indicator of an improved public health system but a determinant of vaccine confidence. Fifth, effective communication in vaccine rollout entails respect by not treating vaccine skeptics as ill-informed. Instead, it requires considering the concerns of the vaccine-hesitant. This is because the vaccine-hesitant could change their minds and be open to an inclusive discussion. Inevitably, there will be individuals who will suffer from the side effects of population-based health promotion, but the general public should be able to trust the recommended vaccines since vaccines are among the greatest innovations of all time.

## Figures and Tables

**Figure 1 diseases-11-00032-f001:**
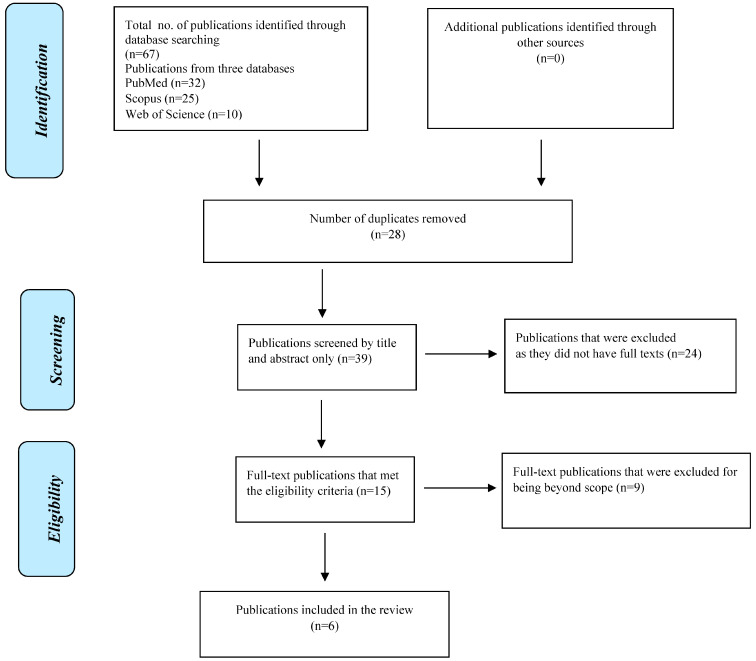
Flow diagram showing the study selection criteria.

**Table 1 diseases-11-00032-t001:** Table showing the eligible papers for analysis of vaccine hesitancy n = 6.

	Article	Search	Selection Criteria	Information Retrieval	Quality Assessment	Major Outcomes
1.	Afolabi and Ilesanmi [13]	PubMed (Journal article—Primary source)	COVID-19 vaccine hesitancy in Africa	Vaccine hesitancy is not a new concept in SSA. There is need for measures to be put in place to prevent COVID-19 vaccine hesitancy, for example, involving the community in the delivery of the vaccines.	The paper advocates for the integration of COVID-19 vaccines into the routine immunization schedule to improve vaccine uptake.	Possible causes of COVID-19 vaccine hesitancy in Africa include public distrust; delayed emergency response in the continent, i.e., laxity in border closures and the subsequent importation of the virus to Africa; and a lack of community involvement in the contexts of social distancing, handwashing and masking, among other measures. African governments also did little to debunk social and traditional media theories that the African continent is “immune” to the virus due to the climatic conditions. As a result of these reasons, many Africans lacked confidence in the manner in which the virus was being reported and handled in the continent.
2.	Ekwebelem et al. [19]	Science Direct Letter to the Editor (Opinion piece)	COVID-19 vaccine and vaccine hesitancy in Africa	In 2019, vaccine hesitancy was listed by WHO as one of the top threats to health and wellbeing. Vaccine hesitancy is driven by cultural, social, historical, political and individual factors such as values, risk perceptions, emotions, knowledge or beliefs.	Africa is a multicultural and diverse continent. Sociocultural complexity has contributed immensely to sporadic vaccine hesitancy. As a result, hesitancy toward COVID-19 vaccines is expected to ultimately vary in different contexts.	There are fears, misinformation and conspiracy theories being spread by social influencers, religious leaders, anti-vaccinists and political leaders that the vaccine is being used to reduce the rising population and also to track people’s lives through microchips.
3.	Flint [20]	JSTOR (Journal article—Primary source)	COVID vaccine trials in Africa	COVID vaccine trials are happening in contexts where there is a history of biomedical experimentation and abuse, such as in the African continent. Vaccine hesitancy in Africa stems from the colonial geography, structural adjustment programs of the 1980s and 1990s, the HIV/AIDs pandemic, clinical trials involving pharmaceutical companies and the influence of iniquities in global health field.	The backdrop of colonialism is important in understanding the resistance to COVID-19 vaccine trials.	There is a need to rethink more equitable relations within global health.
4.	Harrison and Wu [21]	Scopus (Opinion piece)	Vaccine confidence in the time of COVID-19	A need to reexamine whether the COVID-19 pandemic will ease the historical origins of vaccine hesitancy/refusal in sub-Saharan Africa.	A push for vaccine confidence as a means of conceptualizing and responding to the COVID-19 pandemic in a mutual manner.	A call for a public health culture that embraces vaccine safety. The concept of public health programs must be broader than the delivery of the vaccine (biomedical supply chain).
5.	Nihlén [5]	PubMed (Journal article—Primary source)	Vaccine hesitancy and trust	The paper analyzes vaccination policy from an ethical perspective against the backdrop of growing vaccine hesitancy.	The paper looks into examples of vaccination programs such as those for measles and H1N1 and associated side effects that propel hesitancy.	Vaccine skeptics should not be treated as ill-informed or less educated, but their concerns should be addressed respectfully. Furthermore, the public should trust the message and count on the government to take responsibility for individuals affected by vaccine side effects.
6.	Trogen and Pirofski [22]	Web of Science (Commentary)	Understanding vaccine hesitancy in COVID-19	The initial concerns about the scarcity of COVID-19 vaccines increased public demand, but as supply meets demand, vaccine hesitancy is becoming a defining theme of the pandemic.	The paper differentiates vaccine hesitancy from vaccine refusal. Vaccine refusal carries with it deep political, cultural and emotional underpinnings that are difficult to overcome. Individuals in this group are often described as anti-vaxxers (also known as anti-vaccinists).	Overcoming vaccine hesitancy requires a multipronged approach, especially when the vaccine benefits outweigh the risks.

## Data Availability

All analyzed data are included in this article.

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
