# Peer review of "Vaccine Hesitancy in Sub-Saharan Africa in the Context of COVID-19 Vaccination Exercise: A Systematic Review"

_diseases, 2023, doi:10.3390/diseases11010032_

Round 1
Reviewer 1 Report
Summary:
The study of vaccine inequity and vaccine hesitancy is crucial for sustaining collective immunity in vaccine programs., and a review of these studies is highly beneficial to both researchers and the general public. However, additional information should be included in this study as to why there is such a need to focus on SSA with respect to vaccine hesitancy and how it relates - or, perhaps more importantly, does not relate - to what was/is happening in the rest of the world.
Major comments:
For those who are unfamiliar with the situation in Africa, including statistics that demonstrate, for example, the % still unvaccinated in SSA would be highly beneficial in the introduction and discussion, as well as a comparison to the rest of the world.
This study would be considered more informative if the authors could draw links between the biases in the studies and the individual studies’ conclusions – e.g., does it appear after review of multiple studies that skepticism was driven by any particular demographic or geological factor? In other words, is there an additional benefit to pooling these studies into one review that wouldn't have been perceived by the analysis of each study individually? And is this relevant to other parts of the world or is this unique to SSA?
Minor comments:
In the Methods (lines 111-112), the search strategy outlines the use of four key phrases, with the joiner “and.” However, in line 117, the authors state that articles published prior to the year 2000 were removed. I don’t see how this is possible if all four key phrases were considered criteria for inclusion. Perhaps the authors should be referring to “or” when discussing the key phrases?
Line 65 – “concerns over-restrictive legislation” should be “concerns over restrictive legislation”
Lines 73-74 – “a more frequent disease outbreak” should simply be “more frequent disease outbreaks”
In Figure 1, parentheses should be added to the screening square surrounding “n=39” to be consistent.
In line 137 - Should “raising” not be “rising” here?
I think the reader would benefit from an expanded section on study quality (lines 130-139). For example, it is somewhat unclear here what is meant by study biases among the included papers and why this review section should be considered “study quality“ - were the authors wanting to focus on biased studies? Based on the content, the authors appear to be discussing sampling bias, so perhaps study size and location should be mentioned for the studies, e.g. a mean? If not, a clear distinction should be made between sampling bias and opinion/perception bias. The third point, in particular, regarding bias in reporting vaccine hesitancy owing to “weak health systems” should be explained in a little more detail.
In lines 143-146 – “The first paper by Afolabi and Ilesanmi [13] reports that COVID-19 vaccine hesitancy in Africa is fuelled by public distrust resulting from delayed response to preventive activities in the continent; for example, a laxity in border closures may have propelled the importation of the virus to Africa” – the example does not fit the statement. How did lax border closures and propelled introduction result in public distrust? Was the link here that the population deemed the laxity of the government as an indication that relaxation of prevention measures could be taken at the individual level without severe consequence as well? Or was it more that the poor handling of the situation by the government led directly to the distrust?
Similar to the discussion of the first papers, the paper would benefit from the inclusion of examples in the discussion of the other papers.
Missing period line 182.
Line 202 of the discussion – “inequities in global health research” was not really discussed. The varying reasons for hesitancy were discussed, but not in the context of inequities. If the authors want to state that the reviewed papers demonstrate hesitance in light of inequities, these inequities need to be discussed more.
In line 225, the authors of the cited manuscript should be included in the sentence for easy reading.
Line 240 – “COVID19” should be “COVID-19”
Lines 239-24 – “first, the low-resourced nature of many African countries makes it challenging to report on the current state of COVID19 vaccines in the continent.” It would help to understand this sentence if the authors discussed which were actually included in the study and which were not.
Lines 241-242 – “Second is the challenge of finding literature that tackles religious, cultural and other context specific concerns on vaccine information trends.” Why do the authors believe this challenge exists?
I would disagree with lines 269-271 – “Furthermore, developing a COVID-19 vaccine should not be the sole indicator of a successful response nor an indicator of an improved public health system but a better indicator of vaccine confidence.” Vaccine development in and of itself is not an indicator of vaccine confidence.
Line 273 – “vaccinehesitant” should be “vaccine-hesitant”
Author Response
Thank you for the comments; we have revised the manuscript accordingly. Please find the attached document.

Reviewer 2 Report
I recommend rejection of this paper since I think it does not give significant contribution to this field and I don't think it could be appealing for international readers. Regarding the first point, the author should note that the same topic has been recently analyzed in a well done review authored by Deml and Githaiga (Deml MJ, Githaiga JN. Determinants of COVID-19 vaccine hesitancy and uptake in sub-Saharan Africa: a scoping review. BMJ Open. 2022 Nov 18;12(11):e066615. doi: 10.1136/bmjopen-2022-066615. PMID: 36400736; PMCID: PMC9676416). This review covers the same topic and discusses a significantly higher number of papers. The discussion of the submitted draft is in my opinion poor from a scientific point of view and lacks a substantial focus of interest (also with regard of the scope of the journal). Moreover, I strongly recommend that reviews are performed in compliance with PRISMA criteria or other explicit standardized criteria. Regarding the methods, there are some inconsistencies between the numbers in the text and in the table.
Author Response
Thanks for the comments; we have addressed the issues accordingly.

Reviewer 3 Report
The following issues may be considered to resolve, pertaining to the article titled “Vaccine Hesitancy in Sub-Saharan Africa in the Context of 2 COVID-19 Vaccination Exercise: A Systematic Review” by Elizabeth A. Ochola.
Abstract:
1. Please expand the abbreviated term “SSA” to “sub-Saharan Africa”
Introduction:
2. The author may consider merging the section “Objective” with that of “Introduction”.
Last paragraph of Introduction section may be re-constructed as follows:
“Countries in SSA have faced challenges in the COVID-19 vaccine administration. Some of the challenges include poor infrastructure to distribute the vaccine, limited cold chain facilities and a growing vaccine hesitancy [16]. Through platforms like COVID-19 Vaccines Global Access (COVAX), an initiative spearheaded by WHO alongside other partners, ensure equitable access to vaccines globally [17]. Even though COVAX is majorly founded on a mission to accelerate the development and manufacture of COVID-19 vaccines by guaranteeing a fair and equitable distribution, vaccine hesitancy [17] remains a critical component of vaccine rollout that needs to be addressed in SSA. The scope of this review article is meant to evaluate the vaccine hesitancy in sub-Saharan Africa in the context to COVID-19 vaccination.”
Method:
3. Change the following sentences in line 117-119. There is no tool used in the manuscript. Instead the term “data analysis approach” will be more suitable.
“The data extraction approach utilized taking into consideration the articles and database search, study selection criteria, information retrieval, quality assessment, and major outcomes (Table 1). The approach was aimed at obtaining key details and develop themes for analysis.”
4. Re-construct the 2nd paragraph as follows:
“Initial data extraction approach resulted in identifying 67 relevant publications from the databases (PubMed, Scopus and Web of Science). Out of these, only 39 publications were considered for further analysis based on the title and abstract. Further, 24 duplicate articles were removed; leaving with a final pool of 15 articles considered. Subsequently, due to irrelevancy to the main theme of this current review; 9 publications were excluded from, and the remaining 6 articles were analyzed (Figure 1).”
5. Figure 1 seems to be very vague. Not detailed information a reader can draw about the how the author has screened the 6 publications from this flow diagram. Also, PLEASE TRY TO MAINTAIN CONSISTENCY WHEN USING TERMS MENTIONED IN THE FLOW DIAGRAM WITH THAT OF THE RUNNING TEXT OF THE MANUSCRIPT.
One example is “Studies included in qualitative synthesis (n = 6)”. What does “qualitative synthesis” means? Please try to make the article reader friendy.
6. Figure 1 title “Flow diagram showing the selected” is incomplete. Please change this to “Flow diagram showing the study selection criteria” or something that makes more sense.
7. Table 1 is very hard to read. Please align it properly. Preferably use left alignment for all the contents of the Table 1 excluding the Title row of the table which is rightfully placed as “center aligned”. PLEASE CHECK THOROUGHLY FOR FREQUENT GRAMMAR ISSUES THROUGHOUT THE TABLE BEFORE RE-SUBMISSION.
Study quality:
Is this a sub-section of “Methods”? Please check with the manuscript preparation guidelines of the journal whether such an additional section is acceptable or not. If it is not acceptable then make it a sub-section or merge with Method section as an independent paragraph.
Results:
8. The authors tried her best to write the result section. However, this needs to be more elaborated in the revised submission. PLEASE CHECK THOROUGHLY FOR FREQUENT GRAMMAR ISSUES THROUGHOUT THE TABLE BEFORE RE-SUBMISSION.
The paragraphs of the result section seem to be very monotonous when reading. Onme example is the authors is starting almost every paragraph in the result like this “The first/second/third/fourth/fifth/sixth paper by …..”. Please re-structure while writing the section to avoid a monotonous pattern in the article!!
Discussion:
9. Line # 225 “In the quest to remedy vaccine hesitancy, [5] recommends that governments xxxxxx” is wrongly structured. The authors may re-write this as “In the quest to remedy vaccine hesitancy, it has been earlier suggested that that governments xxxxxxxxxx affected by the adverse effect of the vaccines [5].”
Author Response
Thanks for the comments; we have made the corrections as per the attached document.

Round 2
Reviewer 3 Report
The author has tried to improvise the manuscript, but this still needs corrections.
Before re-submission, the author please change the term “records” to “publications” or “record” to “publication” throughout the text in Abstract, Method, Figure-1, and Limitations.
Method:
I have previously commented the following query for Figure 1, which didn’t seem to be resolve as of now. I just gave one example in my previous comment where the author could improvise the figure. There still are issues like they mentioned “Full-text articles assessed for eligibility (n=15)”!!! What “eligibility”?? PLEASE EXPLAIN THE FIGURE IN DETAIL SO THAT ANY READER CAN UNDERSTAND WHAT THE AUTHOR WANT TO SAY FROM A GLIMPSE AT THE FLOW DIAGRAM.
ALSO, PLEASE TRY TO MAINTAIN CONSISTENCY WHEN USING TERMS MENTIONED IN THE FLOW DIAGRAM WITH THAT OF THE RUNNING TEXT OF THE MANUSCRIPT. PLEASE USE THE TERM “PUBLICATIONS” instead of “RECORDS”!!
Previous comment:
5. Figure 1 seems to be very vague. Not detailed information a reader can draw about the how the author has screened the 6 publications from this flow diagram. Also, PLEASE TRY TO MAINTAIN CONSISTENCY WHEN USING TERMS MENTIONED IN THE FLOW DIAGRAM WITH THAT OF THE RUNNING TEXT OF THE MANUSCRIPT.
One example is “Studies included in qualitative synthesis (n = 6)”. What does “qualitative synthesis” means? Please try to make the article reader friendly.
Author Response
Thanks for your kind comments. The diagram has been revised as suggested
